# Semantic Similarity Covariance Matrix Shrinkage

**Guillaume Becquin**
Bloomberg
gbecquin1@bloomberg.net

**Saher Esmeir**
Bloomberg
sesmeir2@bloomberg.net

## Abstract

An accurate estimation of the covariance matrix is a critical component of many applications in finance, including portfolio optimization. The sample covariance suffers from the curse of dimensionality when the number of observations is in the same order or lower than the number of variables. This tends to be the case in portfolio optimization, where a portfolio manager can choose between thousands of stocks using historical daily returns to guide their investment decisions. To address this issue, past works proposed linear covariance shrinkage to regularize the estimated matrix. While effective, the proposed methods relied solely on historical price data and thus ignored company fundamental data. In this work, we propose to utilise semantic similarity derived from textual descriptions or knowledge graphs to improve the covariance estimation. Rather than using the semantic similarity directly as a biased estimator to the covariance, we employ it as a shrinkage target. The resulting covariance estimators leverage both semantic similarity and recent price history, and can be readily adapted to a broad range of financial securities. The effectiveness of the approach is demonstrated for a period including diverse market conditions and compared with the covariance shrinkage prior art.

## 1 Introduction

A wide range of domains rely on the analysis of high-dimensional time-series data. For example, this includes medical systems with MRI denoising (Honnorat and Habes, 2022), radar sensors (Kang et al., 2019) post-processing, physics and chemistry, engineering, neuroscience, speech recognition, and quantitative finance (Ledoit and Wolf, 2020b). These applications often require a reliable estimate of the covariance matrix. In the financial domain, the price of a company's stock is influenced, among other factors, by the fundamental characteristics of the company, news articles, and

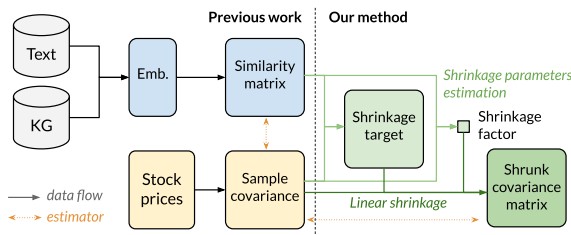

Figure 1: Instead of using the semantic similarity as a direct approximation to the covariance matrix, we use it to build a target for linear shrinkage. This estimator can then be used for convex portfolio optimization.

general stock market conditions. Building upon advances in language modeling and graph embeddings, recent work has investigated how to leverage semantic representations to guide investment decisions by estimating expected returns.

The volatility and the expected return of an asset are both critical ingredients when forming an investment strategy. Risk-return trade-off is a fundamental trading principle describing the typical inverse relationship between a given investment risk and investment return. In the common case of a multi-asset portfolio, an investor with a given risk tolerance seeks to maximise their return within the allowed risk level. In this case, the degree of covariance between asset returns can be utilised to achieve an aggregate portfolio risk that is lower than its components. Modern portfolio theory formalises this trade-off (Markowitz, 1952). The Mean Variance Portfolio aims at maximizing expected returns while minimizing expected variance:

$$\mathbf{w}^* = \underset{\mathbf{w}}{argmax}\left(\mathbf{w}\boldsymbol{\mu} - \lambda\mathbf{w}^T\boldsymbol{\Sigma}\mathbf{w}\right) \qquad (1)$$

where $\mathbf{w}$ denotes the weights of assets in a portfolio, $\boldsymbol{\mu}$ the price returns of these assets, $\boldsymbol{\Sigma}$ the covariance matrix of asset prices and $\lambda$ a parameter to denote the level of risk aversion of the investor. This optimization problem can be solved effectively using convex solvers if the covariance

matrix is positive definite (and therefore invertible). Nevertheless, the covariance matrix of the assets prices is unknown and must therefore be estimated. Kan and Zhou (2007) show that the estimation error for the covariance matrix has a higher impact than the mean estimation error when the number of assets considered is large compared to the number of observations.

Existing approaches leveraging semantic stock representations have focused on improving the estimate of the expected returns $\mu$, using a rough approximation of the covariance matrix $\Sigma$ by directly using the semantic similarity matrix as an estimator for portfolio optimization (Du and Tanaka-Ishii, 2020; Li et al., 2022). However, these similarity matrix estimators may not be morphologically similar to a covariance matrix and are prone to bias. Indeed, while semantic models are effective at ranking similarities, their absolute similarity value is typically uncalibrated and depend on the fraction of the hypersphere occupied by the embeddings. Furthermore, their expected cosine similarity value depends on the dimension of the embeddings (Elekes et al., 2017). Since the true covariance matrix is obviously independent of the chosen embedding dimension, this would imply that a single embedding dimension is valid for building a semantic estimator, which is unacceptable. Finally, they give no guarantee of being positive semi-definite and ignore recent stock price observations.

An alternative estimator for the covariance matrix is the sample covariance matrix. While unbiased, the sample covariance matrix is singular when the number of observations $N$ is lower than the number of random variables $p$ (preventing the use of convex solvers for equation 1) or ill-defined when $N \gg p$, making the portfolio optimization task inaccurate (Ledoit and Wolf, 2004). Covariance shrinkage refers to a family of regularization methods aiming at combining the maximum likelihood sample covariance matrix with a structured, lower-dimensionality, regularizing model (shrinkage target). The regularizing targets are built on a set of assumptions (e.g., *constant variance and 0-covariance between variables*) whose degree of validity varies between asset classes and market regimes. Furthermore, the shrinkage estimators can be interpreted as empirical Bayes estimators where the prior is data-dependent, and computed from the same data as the sample covariance matrix (Ledoit and Wolf, 2020b).

In this work, we highlight the limitations of using semantic similarity as a direct estimation of the covariance matrix. We address these limitations by extending the established framework of linear shrinkage to semantic similarity matrices as seen in Figure 1. The semantic representations can be derived from both structured data stored in a curated knowledge base or unstructured data in the form of natural language text. We share a Python implementation[1] of our method, allowing its integration as a simple post-processing step when building covariance matrix estimators from semantic similarity matrices. We aim at validating the following:

- **Hypothesis 1**: Semantic similarity depends on the embedding dimension and therefore cannot be used directly as a covariance matrix estimator.

- **Hypothesis 2**: Semantic similarity can be used as an effective regularization target for covariance matrix estimation.

- **Hypothesis 3**: The proposed shrinkage target includes fundamental information about the random variables and is therefore less sensitive to sudden changes in volatility regime.

## 2 Related Work

**Portfolio optimization using semantic representations:** Du and Tanaka-Ishii (2020) investigate the potential of using news data to train a classifier to make buy and sell decisions over time. They take the cosine similarity matrix of intermediate stock representations as a direct estimator for the covariance matrix to perform portfolio optimization. Similarly, Li et al. (2022) use normalized stock embeddings trained on fundamental and price data to build a similarity matrix and use it as an estimate of the covariance matrix for mean-variance portfolio optimization. Both approaches suffer from the assumption that the similarity matrix is an unbiased estimator of the covariance. Our work instead uses semantic similarity as a shrinkage target to the sample covariance matrix resulting in estimators that are asymptotically unbiased. Sawhney et al. (2021) leverage a knowledge graph (KG) to model the correlation between stocks through temporal hyperbolic graph learning on Riemannian manifolds. They train a model to directly output a ranked list

---

[1] https://github.com/bloomberg/semantic-similarity-covariance-shrinkage

of trade candidates, and therefore do not rely on the Modern Portfolio Theory mean-variance optimization problem. Our approach instead leverages established portfolio optimization routines allowing better interpretability and modularity (decoupling the semantic representations generation from the investment strategy).

**Covariance shrinkage:** Linear shrinkage methods, taking a convex linear combination between the sample covariance matrix and a regularization target were originally proposed by Stein (1956) for the estimation of high dimensional multivariate mean, choosing the 0-vector as a shrinkage target. Efron and Morris (1973) built upon this approach by choosing a 1-parameter structured target built as the identity vector multiplied by the sample mean. This work was extended to a robust estimation of the covariance matrix in a high dimensional space by Ledoit and Wolf (2004), proposing a single-parameter shrinkage target built as the scalar product of the identity matrix and the average sample variance. Their choice of shrinkage target made the strong assumption that the stock price returns had a constant, common variance and no covariance. Ledoit and Wolf (2003a) instead assumed a constant and positive correlation between stock price returns, resulting in a 2-parameters shrinkage target. The single-factor model (Ledoit and Wolf, 2003b) assumes the asset prices can be approximated using the capital asset pricing model (CAPM) and linearly depend on the average market returns with sensitivity $\beta_i$ and an offset $\alpha_i$ capturing the stock idiosyncratic returns. The resulting shrinkage target contains $p$ parameters to estimate, higher than the previous two models, but typically significantly lower than the $p(p-1)/2$ parameters of the full (symmetric) covariance matrix. These regularization targets are estimated using the same data used to compute the sample covariance matrix and are therefore prone to estimation error if the number of samples is low relative to their number of parameters. Our approach instead utilises similarity-based targets that are not derived from recent observations but rather trained on large scale semantic datasets. These therefore do not suffer from this parametrization trade-off and are less sensitive to market regime changes.

## 3 Preliminaries

In this section, we formally describe the framework of linear shrinkage and derive the optimum shrink-

age (regularization) factor coefficient. Let $X_N$ be a de-meaned price returns matrix of $N$ observations for $p$ assets (we omit the $p$ for notation simplicity). The sample covariance matrix is a maximum likelihood estimator (MLE) of the covariance and is defined as:

$$S_N = \frac{1}{N} X_N^T X_N \tag{2}$$

The sample covariance matrix is symmetric and has $\frac{p(p-1)}{2}$ parameters to estimate. When the number of observations $N$ is lower than the number of features $p$, this matrix is non-invertible. Furthermore, unless $p \ll N$, it is numerically ill-conditioned and the instabilities in the estimate lead to the underestimation of the smallest eigenvalues and overestimation of the largest ones (Ledoit and Wolf, 2004). An alternative is to replace this MLE by a lower capacity model making a series of assumptions on the random variables observed (for example constant variance, no correlation or a factor model as introduced previously). This option is also unsatisfactory as it would suffer from a high bias because of the strong modeling assumptions implied. The approach of linear shrinkage is to build a new estimator from a convex linear combination of (1) the high variance, low bias of the sample covariance matrix; and (2) the low variance, high bias of the shrinkage target (lower capacity model) $T$:

$$\hat{\Sigma}_N = \lambda T + (1 - \lambda) S_N$$
$$\lambda \in [0, 1] \tag{3}$$

The objective is to minimize the mean square error between this estimator and the population covariance $\Sigma^*$. The loss function is therefore given by

$$\begin{aligned}\mathcal{L}_\lambda &= \|\hat{\Sigma}_N - \Sigma^*\|_F^2 \\ &= \|\lambda T + (1 - \lambda) S_N - \Sigma^*\|_F^2 \end{aligned} \tag{4}$$

The expected risk is

$$\begin{aligned}R(\lambda) &= \mathbb{E}(\mathcal{L}_\lambda) \\ &= \mathbb{E}(\|\lambda T + (1 - \lambda) S_N - \Sigma^*\|_F^2) \end{aligned} \tag{5}$$

The population covariance is unknown and not observable but Ledoit and Wolf (2003b) show that the optimum $\lambda^*$ coefficient is asymptotically con-

stant over $N$ and has the following form:

$$\lambda_N^* = \frac{1}{N}\frac{\pi - \rho}{\gamma} \tag{6}$$

$$\pi_N = \sum_{i=1}^{p}\sum_{j=1}^{p} AsyVar(\sqrt{N}S_{ij}) \tag{7}$$

$$\rho_N = \sum_{i=1}^{p}\sum_{j=1}^{p} AsyCov(\sqrt{N}T_{ij}, \sqrt{N}S_{ij}) \tag{8}$$

$$\gamma_N = \sum_{i=1}^{p}\sum_{j=1}^{p} (T_{ij} - S_{ij})^2 \tag{9}$$

where $AsyVar$ and $AsyCov$ respectively refer to the asymptotic variance and covariance when the number of observations $N$ grows larger.

Intuitively, $\pi_N$ measures the degree of variance in the sample covariance matrix, with high values indicating high variance for the sample covariance estimator. A high $\pi_N$ would therefore result in a higher amount of required shrinkage. $\rho_N$ measures the level of covariance between the sample covariance matrix and the shrinkage target: a high covariance indicates the chosen target provides limited additional information and therefore the optimum shrinkage amount should be reduced (Ledoit and Wolf, 2020b). Finally, $\gamma_N$ measures the distance between the unbiased sample covariance matrix and the target. A higher value indicates a biased shrinkage target, and therefore leads to a lower optimum shrinkage factor.

## 4 Semantic Similarity Shrinkage

Our proposed approach is based on the availability of embeddings (vector representation) for the random variables considered. In the context of financial markets, this translates into the availability of a $(p, k)$ embeddings matrix $\mathbf{E}$ built from $k$-dimensional semantic vectors for the $p$ financial assets considered for the portfolio optimization. These embeddings may be generated using text description of the assets (e.g., description of the company that issued a stock) and a sentence embeddings model such as (Reimers and Gurevych, 2019), KG embeddings, or intermediate representation generated from a downstream task such as a buy/sell classifier (Du and Tanaka-Ishii, 2020). We first propose a semantic shrinkage target built using these embeddings and then derive the optimum shrinkage factor.

### 4.1 Semantic shrinkage target

Our approach for building a shrinkage target from embeddings matrices is as follows. We first normalize the embeddings matrix $\mathbf{E}$ along the semantic dimension, bounding the dot product between two asset representations in $[-1; 1]$. We then build a $(p, p)$ similarity matrix $\mathbf{Sim}$ by taking the product of the embeddings matrix with itself.

$$\mathbf{Sim} = \mathbf{E}\mathbf{E}^T \tag{10}$$

This results in a matrix where the diagonal entries are 1 while each off-diagonal entry $(i, j)$ represents the degree of semantic match between assets $i$ and $j$. Rather than directly considering $\mathbf{Sim}$ an approximation of the covariance matrix as in (Li et al., 2022), we instead consider $\mathbf{Sim}$ to be an approximation of a correlation matrix (based on its constant $1-$valued diagonal and entries bounded in $[-1; 1]$). We map this semantic correlation to a covariance matrix using the covariance definition and the asset variance estimated from the sample covariance matrix. The correlation between two random variables $x$ and $y$ is defined as:

$$corr(x, y) := \frac{cov(x, y)}{\sqrt{var(x)var(y)}} \tag{11}$$

By analogy, we define the similarity-based covariance as:

$$\mathbf{Sim} := \frac{\mathbf{SimCov}}{\sqrt{\mathbf{diag}(S_N)^T\mathbf{diag}(S_N)}} \tag{12}$$

$$\mathbf{SimCov} = \mathbf{Sim} \odot \sqrt{\mathbf{diag}(S_N)^T\mathbf{diag}(S_N)} \tag{13}$$

where $\odot$ is the Hadamard product and $\mathbf{diag}(S_N)$ is a $(1, p)$ vector made of the diagonal entries of the sample covariance matrix $S_N$. This approach ensures that the scale of the semantic-based shrinkage target is on the same scale of the sample covariance matrix, regardless of the original stock embeddings space. This estimator therefore has $p$ price-dependent parameters to estimate (the variances), similar to the CAPM-based single factor model (Ledoit and Wolf, 2003b).

Finally, we ensure that the shrinkage target is positive definite, a desirable property that allows the use of the shrunk covariance matrix with convex optimizers. While the covariance matrix $\mathbf{SimCov}$ is by construction symmetric, it is not necessarily positive definite. We introduce small perturbations

on the eigenvalues following spectral decomposition of **SimCov** based on (Gill et al., 2019). Let $U$ and $\Delta$ be the eigenvectors and eigenvalues from the the spectral decomposition of **SimCov**:

$$\mathbf{SimCov} = U\Delta U^T \tag{14}$$

If $min(\Delta) < 0$, we shift the eigenvalues so that they are all strictly positive with a margin $\epsilon$:

$$\Delta_+ = \Delta + min(abs(\Delta)) + \epsilon \tag{15}$$

We then recover the semi-definite similarity-based covariance shrinkage target:

$$\mathbf{SimCov}_+ = U\Delta_+ U^T \tag{16}$$

We do not use this value directly to approximate the covariance matrix; instead we build a linear shrinkage estimator from the convex linear combination of the target and sample covariance matrix as per equation 3:

$$\hat{\Sigma}_{Sim} = \lambda\mathbf{SimCov}_+ + (1-\lambda)S_N \tag{17}$$

## 4.2 Semantic shrinkage factor

As mentioned in Section 3, the optimum shrinkage factor for the similarity-based covariance estimator can be derived analytically using equations 7 to 9. The following provides consistent estimators when assuming a semantic similarity target. Index $t$ refers to time, while $i$ and $j$ refer to the position of the random variables in the covariance matrix. The derivation is similar to constant correlation shrinkage (using similarity values instead of a constant value) and is provided in Appendix A.

A consistent estimator of $\pi_N$ is given by (Ledoit and Wolf, 2020b):

$$\hat{\pi}_N \quad = \sum_{i=1}^{p}\sum_{j=1}^{p} \pi_{\hat{N},ij} \tag{18}$$

$$\hat{\pi}_{N,ij} = \frac{1}{N}\sum_{t=1}^{N}(x_{ti}x_{tj} - S_{N,ij})^2$$

$$= \frac{1}{N}X2_N X2_N^T - S_{N,ij}^2 \tag{19}$$

with $X2_N = (X_N - \bar{X}_N)^{\odot 2}$ the matrix of element-wise squared de-meaned returns.

We obtain a consistent estimator $\hat{\rho}_N$ based on the constant variance target proposed in (Ledoit and Wolf, 2003a), where the constant correlation value is replaced by entries in the similarity matrix. We use the fact that the similarity matrix is constant

with respect to the observations and symmetric, a full derivation is available in Appendix A.

$$\hat{\rho}_N = \sum_{i=1}^{p} \hat{\pi}_{N,ii} +$$
$$\sum_{i\neq j}^{p} \frac{e_{ij}}{2}\left(\sqrt{\frac{s_{jj}}{s_{ii}}}\hat{\vartheta}_{ii,ij} + \sqrt{\frac{s_{ii}}{s_{jj}}}\hat{\vartheta}_{jj,ij}\right) \tag{20}$$

Where the $\hat{\vartheta}_{ii,ij}$ are consistent estimators of $AsyCov(\sqrt{N}S_{N,ii}, \sqrt{N}S_{N,ij})$ given by:

$$\hat{\vartheta}_{ii,ij} = \frac{1}{N}\sum_{t=1}^{N}(x_{ti}^2 - S_{N,ii})(x_{ti}x_{tj} - S_{N,ij}) \tag{21}$$

Finally, a consistent estimator of $\gamma_N$ is given by:

$$\hat{\gamma}_N = \sum_{i=1}^{p}\sum_{j=1}^{p}(\mathbf{SimCov}_{+i,j} - S_{N,i,j})^2 \tag{22}$$

$$= \|\mathbf{SimCov}_{+i,j} - S_{i,j}\|_F^2 \tag{23}$$

This fully specifies the optimal shrinkage factor based on 6.

## 5 Experiments

### 5.1 Datasets

We evaluate the performance of the proposed similarity-based covariance matrix shrinkage on the task of portfolio selection. In particular, we consider the universe of historical members for 3 major stock indices: the Standard & Poor's 500 (S&P 500), the NASDAQ-100 and the Nikkei 225 (Nikkei). We choose this data based on the availability of price data and high quality company fundamental data that can be used to generate stock embeddings. We report results for the S&P 500 for conciseness; the full set of results for all 3 indices is available in Appendices C and D.

#### 5.1.1 Stock price return data

We gather price data over a period of 16 years from 2007 to 2023. This extended period includes a variety of market regimes including high volatility events, such as the global financial crisis in 2007-2008, the sovereign debt crisis in 2011, and the coronavirus pandemic in 2020. It also includes periods of low volatility such as 2017, when the volatility index (VIX) was at historic lows. We build the dataset as follows:

1. We split the data in rolling windows of 3 months for estimator fitting and 1 month for evaluation.

2. The stock indices composition evolved over the period considered. Within each window, we keep only index members with price data throughout the entire period.

3. We convert the end-of-day price into daily price returns, removing the effect of the absolute stock price.

This results in 190 slices, each covering a 4 month span (the first 3 months are used for fitting and the last for evaluation).

### 5.1.2 Semantic representations

We compare the effectiveness of the proposed approach using three methods for computing stock semantic representations. The set of hyperparameters used for the training of the semantic models is available in Appendix E.

The first method leverages the text description for the companies that issued stock. We build semantic representation for the companies using a 768 dimensions sentence embeddings transformer model (Reimers and Gurevych, 2019). The experiments were conducted using a DistilBERT model (Sanh et al., 2019) finetuned on the Natural Questions dataset (Kwiatkowski et al., 2019). The second method leverages relational facts from knowledge graphs. We use publicly available 512 dimensions RotatE pre-trained embeddings on the Wikidata5m dataset generated using GraphVite (Zhu et al., 2019), referred to as **W5m KG**. For both methods we match the stock price data to a Wikidata entry using the Wikidata5m dataset (Wang et al., 2021) containing entity names and aliases. We successfully match 98.3%, 97.9% and 99.3% of index members for the S&P 500, NASDAQ-100 and Nikkei 225, respectively. The unmatched entries are excluded from both training and evaluation rolling windows.

We then consider a third model using a subgraph from a large scale, financial knowledge graph in Bloomberg herein referred to as **Fin. KG**. This includes triples involving the domicile country, industry, sector, board members, supply chain, subsidiaries, fund and index inclusion. We train KG embeddings using RotatE (Sun et al., 2019) with a 128 dimensions embeddings size.

We observe that the representations produced by these methods capture the overall semantics of the company as illustrated in Figure 2. We build the semantic similarity matrices for all semantic models following the method described in 4.1.

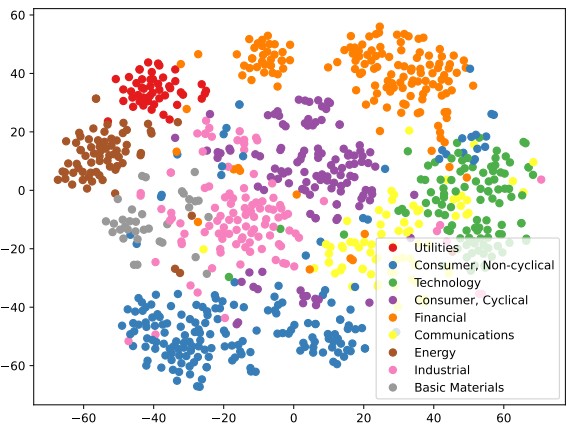

Figure 2: 2-D projection of the KG embeddings using t-SNE. Each point corresponds to a historical S&P 500 component colored by BICS industry sectors.

### 5.2 Performance of semantic-based shrinkage targets

We build the covariance matrix for each evaluation period in 5.1.1 and annualize the results by multiplying the entries by the number of trading days per year (252). For each of the 3-month training periods (first 3 months), we fit the following covariance estimators:

- Sample covariance matrix $S_{Sample}$.

- Constant-variance shrunk covariance matrix $\hat{\Sigma}_{CVar}$ using a 1-parameter shrinkage target assuming a constant variance and no covariance (Ledoit and Wolf, 2004).

- Constant-correlation shrunk covariance matrix $\hat{\Sigma}_{CCor}$ using a 2-parameters shrinkage target assuming a constant correlation (Ledoit and Wolf, 2003a).

- Single-factor shrunk covariance matrix $\hat{\Sigma}_{SF}$ assuming a single factor CAPM model for the stock price returns (Ledoit and Wolf, 2003b).

- Scaled Text similarities $\hat{\Sigma}_{Text,scaled}$ using the Wikipedia text embeddings similarity matrix transformed using 13.

- Scaled KG similarities $\hat{\Sigma}_{W5m\ KG,scaled}$ and $\hat{\Sigma}_{Fin.\ KG,scaled}$ using the KG similarity matrices transformed using 13.

- Shrunk covariance matrix $\hat{\Sigma}_{Text}$, using the Wikipedia text embeddings similarity matrix.

- Shrunk covariance matrices $\hat{\Sigma}_{W5m\ KG}$ and $\hat{\Sigma}_{Fin.\ KG}$ using the KG similarity matrices.

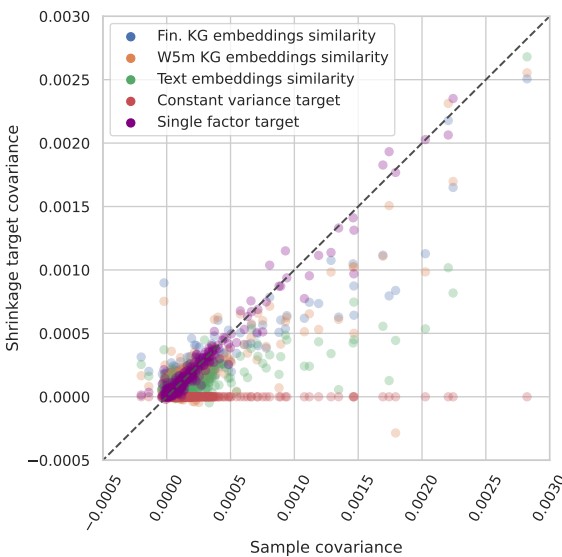

Figure 3: Off-diagonal entries of shrinkage targets relative to the sample covariance.

| Estimator | Mean $\|.\|_F$ |
|-----------|------------|
| Sample covariance | 0.128 |
| Text Sim. | 0.066 |
| W5m KG Sim. | 0.101 |
| Fin. KG Sim. | 0.114 |

Table 1: Average off-diagonal Frobenius norm

We begin by observing the impact of the shrinkage model assumptions on the resulting target bias. We sample 500 ($time, stock\ pair$) observations for all estimators for the S&P 500 and compare their value to the corresponding entry in the sample covariance matrix for the same ($time, stock\ pair$) (see Figure 3). Values along the diagonal indicate a shrinkage target with no bias. By construction the constant variance model has a 0 value on the off-diagonal and is therefore the most biased estimator considered. The semantic estimators show a variable level of bias. The average Frobenius norm of the similarity estimators (excluding the diagonal variances) is given in table 1. The norm of the unbiased sample covariance matrix is given for reference. It shows that different semantic models may result in significant differences in similarity values and therefore are prone to bias when estimating the covariance, validating **hypothesis 1**.

In order to evaluate the performance of the shrinkage targets, we compute the optimum shrinkage factor based on 4.2 and shrink the sample covariance matrix for each 3-month fitting period to build an estimator. We evaluate these against the

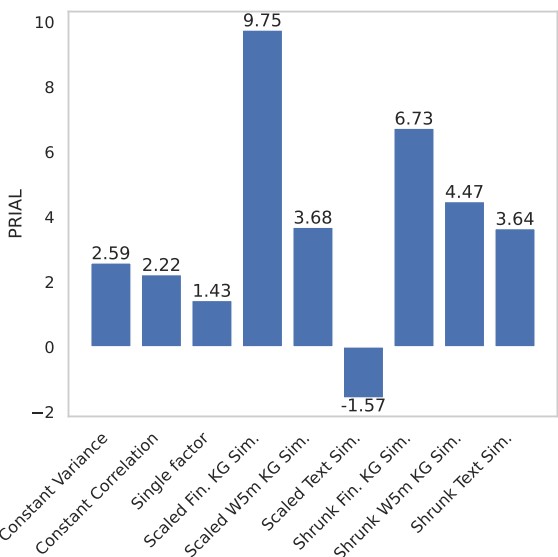

Figure 4: PRIAL for the S&P 500 for 2007-2023

following 1-month sample test covariance matrix as an approximation of the population covariance. We evaluate a given estimator $\hat{\Sigma}$ using the Percentage Relative Improvement in Average Loss (PRIAL) defined as (Ledoit and Wolf, 2020a):

$$PRIAL(\hat{\Sigma}) := 100\frac{\mathbb{E}(\mathcal{L}(S_{Sample})) - \mathbb{E}(\mathcal{L}(\hat{\Sigma}))}{\mathbb{E}(\mathcal{L}(S_{Sample}))} \tag{24}$$

$\mathcal{L}$ refers to the MSE loss introduced in equation 4.

Figure 4 shows the PRIAL for all estimators over the entire period (190 data points) for S&P 500. The constant variance model is the best performing price-based shrinkage target. Using the scaled similarity matrix shows a high level of performance variability and shows the impact of the bias for the text-based embeddings (resulting in a worse estimator than the sample covariance matrix). The higher performance of the scaled KG similarity is highly snapshot-dependent and driven by high volatility events during the period considered (see Appendix B). Our shrunk similarity-based models are outperforming price-based shrinkage methods and their scaled variant. Figure 5 shows the outperformance of shrunk similarity-based methods is consistent across time windows and not driven by isolated high volatility outliers. We perform a statistical test of the null hypothesis (*the performance of similarity-based models and constant variance shrinkage method is equal*) that is rejected with p-values of $1.3 \times 10^{-12}$ and $2.5 \times 10^{-10}$ for the KG-based and text-based similarity targets respectively, validating **hypothesis 2**.

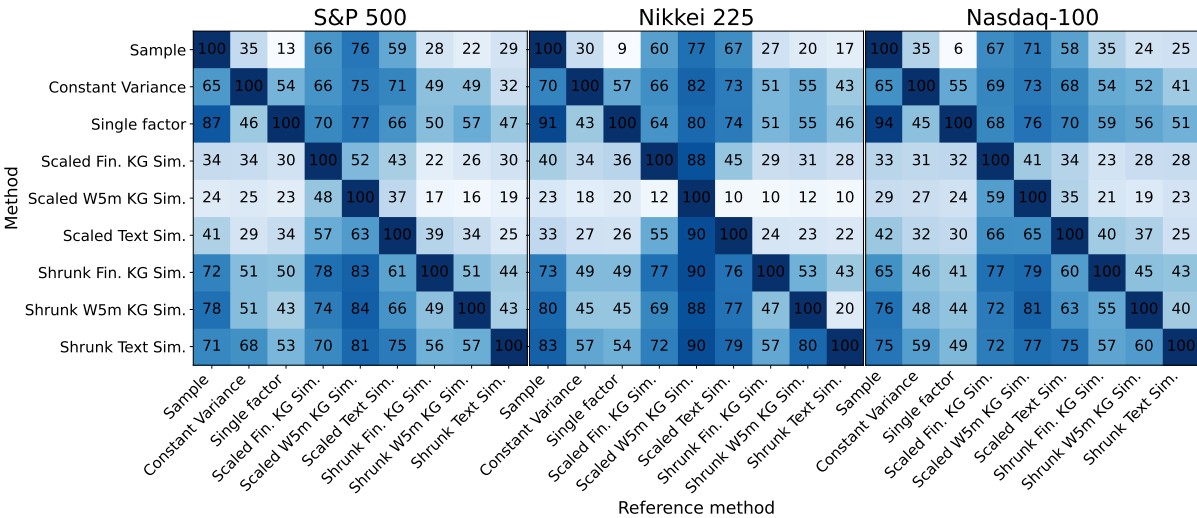

Figure 5: Percentage of rolling windows when a method (left) MSE is lower than a reference method (bottom). The semantic shrinkage methods error is consistently lower than their scaled variant and estimators using price data only.

### 5.3 Impact of volatility regimes on shrinkage target performance

We further investigate the impact of market regime change. In order to identify the contribution of specific samples to the, we define the Percentage Relative Improvement in Loss (PRIL) based on 24:

$$PRIL(\hat{\Sigma}) := 100 \frac{\mathcal{L}(S_{Sample}) - \mathcal{L}(\hat{\Sigma})}{\mathcal{L}(S_{Sample})} \quad (25)$$

Figure 6 focuses on years 2017 to 2023 for the S&P 500 during which several market regimes were observed, including: low volatility (2017 to the end of 2018); two high volatility periods (end of 2018, 2022); and a very high volatility event (first half of 2020). We show PRIL values for both the KG-based similarity and constant variance shrunk covariance models together with the realized annualized log market volatility ($log(\|S_{Sample}\|_F * \#_{trading\ days})$) on the right axis. The outperformance of the similarity-based estimator is higher during periods (or transitions between periods) of high market volatility.

We compute the PRIL difference between the KG-based similarity and constant variance shrunk covariance models (shaded area in Figure 6) for all test periods between 2007 and 2023 to validate this effect, and present the results in Figure 7. We note that the degree of outperformance of the similarity-based approach correlates with the level of market volatility. The effect is particularly pronounced for the very high volatility levels, indicating the potential of the approach to mitigate market "fat

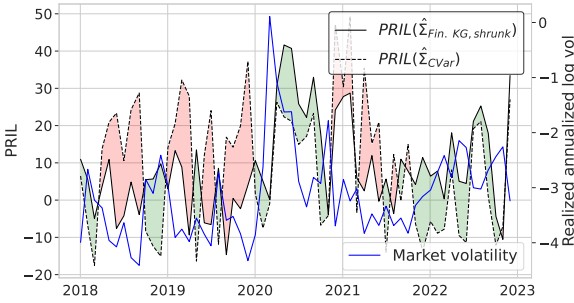

Figure 6: S&P 500 PRIL for the KG-based similarity and constant variance shrinkage targets. The shaded area shows the difference between the PRIL values. An indicator of market volatility is shown on the right axis.

tail risk" (short-term move of more than 3 standard deviations). This validates the **Hypothesis 3** and shows that the proposed covariance matrix estimators are more robust during transitions between market volatility regimes.

## 6 Discussion

In this work, we present a family of covariance shrinkage estimators leveraging semantic similarity between the random variables. We show that semantic similarity cannot be used as a direct estimator of the covariance, but instead as regularization for the sample covariance using linear shrinkage. The shrunk estimators include price-independent semantics and are superior in periods of high volatility, where an accurate estimation of the covariance is typically more critical. This demonstrates the potential of semantic models for covariance matrix estimation and extends beyond

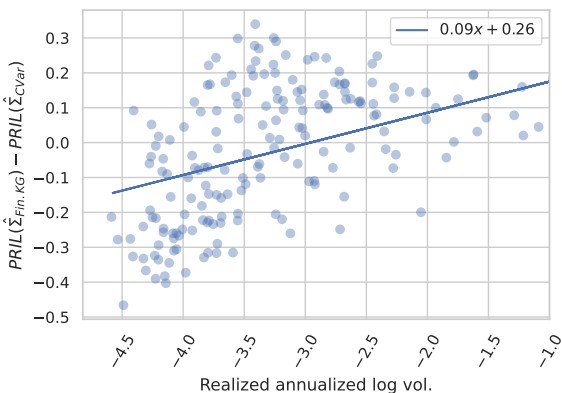

Figure 7: PRIL difference between KG and constant variance shrunk estimators: performance benefits increase with the overall market volatility

traditional applications to predicting returns.

The proposed method is agnostic to the embeddings generation process and can be applied to most existing semantic representation algorithms. This includes semantic representations that capture static (company description) and dynamic (company news, recent price data) components from text, knowledge graphs or multi-modal datasets. By decoupling the semantic model from the portfolio optimization, the proposed method provides a robust and modular mechanism for integrating semantic representations into established portfolio management and investment decision systems.

## 7   Limitations

While showing encouraging results, the proposed similarity-based targets are built from static semantic information. A potential future work direction could be the extension of these static embeddings with dynamic components, including for example news data or dynamic knowledge graphs. Text embeddings from news data (capturing company recent events) can be combined with the static embeddings (capturing company fundamentals) to compute either a single similarity matrix target for shrinkage, or use techniques such as multiple targets shrinkage. Building semantic representations from news data will involve challenges related to sampling for companies with high news coverage, the lack of news for some companies and the choice of importance given to the static versus dynamic components.

The input to the semantic models (text and KG triples) was gathered as of 2021 and 2022, potentially leading to data leakage for the 2007-2023

period backtest. Future work may use historical Wikipedia snapshots to measure the impact of this assumption. However, we estimate this risk to be low given the focus on company fundamentals. The price data frequency is typically much higher than frequency of change of the fundamental data (description or KG triples): the company characteristics (e.g., industry sector) of the large capitalization investigated did not change significantly over the period considered. Furthermore, the analysis does not show a drop in performance before and after the datasets snapshot date. Future work including dynamic semantic components (e.g., news) should however ensure that only news released prior to an experiment time window are used, since news do have a significant impact on short term price data (and correlation) as illustrated by recent work referenced in our work. Similarly, the rolling window evaluation excludes companies that stopped trading during the period considered: this inclusion/exclusion process is a-priori unknown.

The approach requires both rich semantic representations and an effective way to match the stock price returns to their embeddings. The matching process may be more challenging for international companies or smaller capitalization stocks than for the S&P 500, NASDAQ-100 or Nikkei 225 components. Finally, while the approach could be extended to any financial asset class including commodities or derivatives, obtaining a valid text description may prove more challenging. Note that the knowledge-graph embeddings do not suffer from this limitation and can generalize to any asset class where structured data is available, and future work may investigate this method potential for multi-asset portfolio (i.e., equities, fixed income and commodities) covariance estimation.

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

## A Semantic Target Shrinkage Factor Estimation

For notation simplicity we omit in the following the $N$ index from the sample covariance matrix.

### A.1 Derivation of the $\hat{\pi}_N$ estimator

A consistent estimator of $\pi_N$ is given by (Ledoit and Wolf, 2003b):

$$\hat{\pi}_N = \sum_{i=1}^{p} \sum_{j=1}^{p} \pi_{\hat{N},ij} \tag{26}$$

$$\hat{\pi}_{N,ij} = \frac{1}{N} \sum_{t=1}^{N} (x_{ti}x_{tj} - S_{ij})^2 \tag{27}$$

$\hat{\pi}_{N,ij}$ can be rewritten to allow for factorized computation, using the fact that $S_{N,ij}$ is constant with respect to the sum over the samples:

$$\hat{\pi}_{N,ij} = \frac{1}{N} \sum_{t=1}^{N} (x_{ti}x_{tj} - S_{ij})^2$$

$$= \frac{1}{N} \sum_{t=1}^{N} x_{ti}^2 x_{tj}^2 - 2S_{ij} \sum_{t=1}^{N} \frac{x_{ti}x_{tj}}{N} - S_{ij}^2$$

$$= \frac{1}{N} \sum_{t=1}^{N} x_{ti}^2 x_{tj}^2 - 2S_{ij}S_{ij} - S_{ij}^2$$

$$= \frac{1}{N} \sum_{t=1}^{N} x_{ti}^2 x_{tj}^2 - S_{ij}^2$$

$$= \frac{1}{N} X2_N X2_N^T - S_{ij}^2 \tag{28}$$

with $X2_N = (X_N - \bar{X}_N)^{\odot 2}$ the matrix of element-wise squared de-meaned returns.

### A.2 Derivation of the $\hat{\rho}_N$ estimator

We follow the derivation an estimator for $8_N$ assuming a shrinkage target built on a constant correlation matrix provided in (Ledoit and Wolf, 2003a), replacing the constant correlation value $\bar{r}_N$ by individual semantic similarities matrix entries $e_{ij}$. By definition (see 8):

$$\rho_N = \sum_{i=1}^{p} \sum_{j=1}^{p} AsyCov(\sqrt{N}T_{ij}, \sqrt{N}S_{ij}) \tag{29}$$

$$= \sum_{i=1}^{p} AsyVar(\sqrt{N}S_{ii})$$

$$+ \sum_{i \neq j}^{p} AsyCov(\sqrt{N}e_{ij}\sqrt{S_{ii}S_{jj}}, \sqrt{N}S_{ij}) \tag{30}$$

The first term is equal to the diagonal of the $\hat{\pi}_{N,ij}$ estimator calculated previously. As per (Ledoit and Wolf, 2003a), using the delta-method an estimator for $AsyCov(\sqrt{N}e_{ij}\sqrt{S_{ii}S_{jj}}, \sqrt{N}S_{ij})$ is given by:

$$\frac{1}{2} \left( \begin{array}{c} \sqrt{\dfrac{S_{jj}}{S_{ii}}} AsyCov(\sqrt{N}e_{ij}S_{ii}, \sqrt{N}S_{ij}) \\ + \sqrt{\dfrac{S_{ii}}{S_{jj}}} AsyCov(\sqrt{N}e_{ji}S_{ii}, \sqrt{N}S_{ij}) \end{array} \right) \tag{31}$$

Since the $e_{ij}$ are constant with respect to the observations and symmetric ($e_{ij} = e_{ji}$), this can be re-written:

$$\frac{e_{ij}}{2} \left( \begin{array}{c} \sqrt{\dfrac{S_{jj}}{S_{ii}}} AsyCov(\sqrt{N}S_{ii}, \sqrt{N}S_{ij}) \\ + \sqrt{\dfrac{S_{ii}}{S_{jj}}} AsyCov(\sqrt{N}S_{ii}, \sqrt{N}S_{ij}) \end{array} \right) \tag{32}$$

A consistent estimator for $AsyCov(\sqrt{N}S_{ii}, \sqrt{N}S_{ij})$ is given by (Ledoit and Wolf, 2003a):

$$\hat{\vartheta}_{ii,ij} = \frac{1}{N} \sum_{t=1}^{N} (x_{ti}^2 - S_{ii})(x_{ti}x_{tj} - S_{ij}) \tag{33}$$

And by symmetry:

$$\hat{\vartheta}_{jj,ij} = \frac{1}{N} \sum_{t=1}^{N} (x_{tj}^2 - S_{jj})(x_{tj}x_{ti} - S_{ij}) \tag{34}$$

The resulting consistent estimator for $\rho_N$ is given by:

$$\hat{\rho}_N = \mathbf{diag}(\hat{\boldsymbol{\pi}}_N)$$
$$+ \sum_{i \neq j}^{p} \frac{e_{ij}}{2} \left( \sqrt{\frac{s_{jj}}{s_{ii}}} \hat{\vartheta}_{ii,ij} + \sqrt{\frac{s_{ii}}{s_{jj}}} \hat{\vartheta}_{jj,ij} \right) \tag{35}$$

# B Comparison of Scaled and Shrunk Estimators for 2018-2023 period

This appendix provides additional experimental results for the similarity-based estimators for the 2018 to 2023 period. Figure 9 shows that while the Percentage Relative Improvement in Average Loss is higher for a naive estimator built using the scaled similarity as a covariance estimator, the outperformance is concentrated in a single period of very high volatility. Since the loss is a mean squared error, the large covariance values (and their error) seen during this period dominate the general evaluation period.

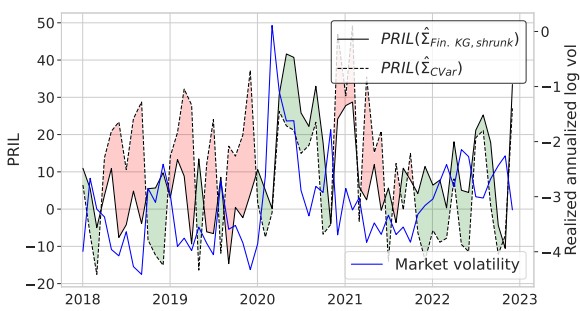

Figure 8: Relative performance of the Fin. KG similarity shrunk estimator to the constant variance estimator

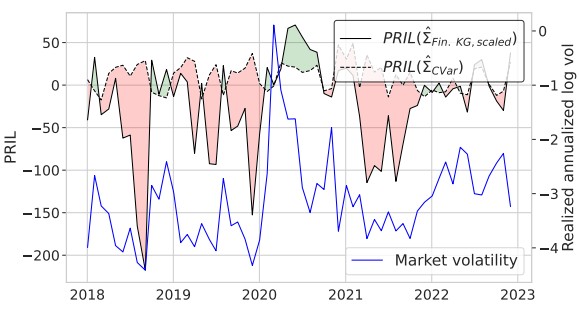

Figure 9: Relative performance of the Fin. KG similarity scaled estimator to the constant variance estimator

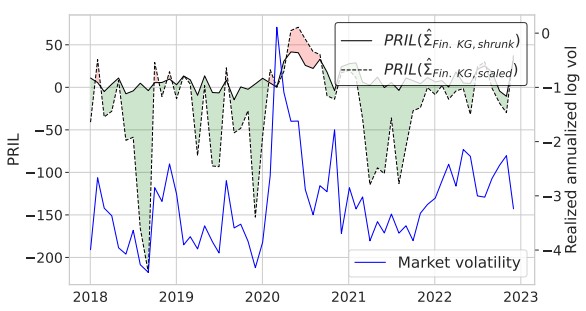

Figure 10: Relative performance of the Fin. KG similarity shrunk estimator to the KG similarity scaled estimator

It can be seen that the estimator under performs both the constant variance shrunk estimator (Figure 9) and the shrunk covariance using a KG similarity matrix target (Figure 10). A similar effect for the text encoder semantic similarity can be seen in Figure 11 to Figure 13, showing the systematic underperformance of uncalibrated semantic similarity matrices for covariance estimation.

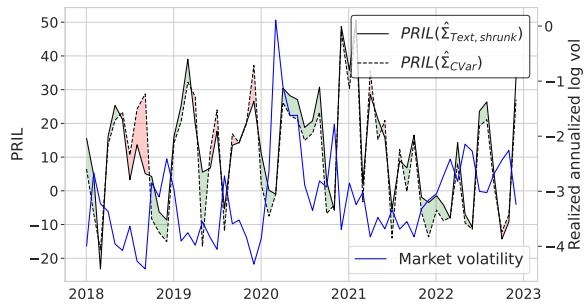

Figure 11: Relative performance of the Text similarity shrunk estimator to the constant variance estimator

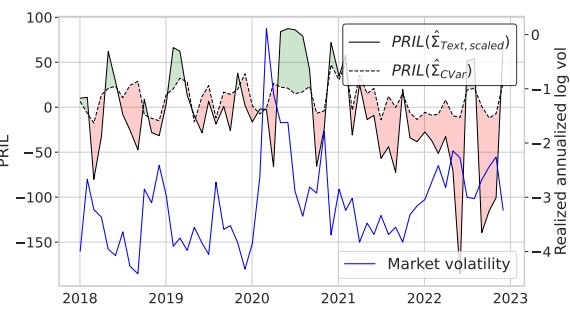

Figure 12: Relative performance of the Text similarity scaled estimator to the constant variance estimator

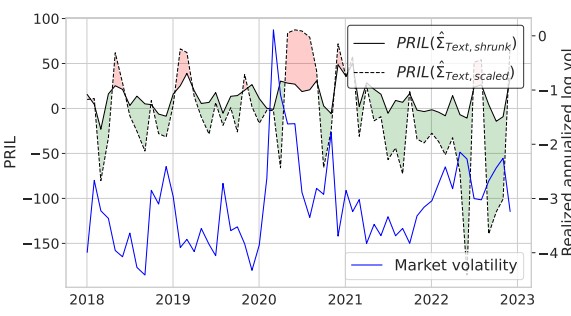

Figure 13: Relative performance of the Text similarity shrunk estimator to the Text similarity scaled estimator

# C  Experimental results for NASDAQ-100

This appendix provides provides the experimental results for the NASDAQ-100 components. Figure 14 shows the PRIAL (defined in equation 24) over the entire experimental period. The results align with the S&P 500, showing a strong variance in the average performance of scaled semantic estimators (with the text semantic similarity under-performing the sample covariance estimator). This variance is significantly lower for the shrunk estimators (our method).

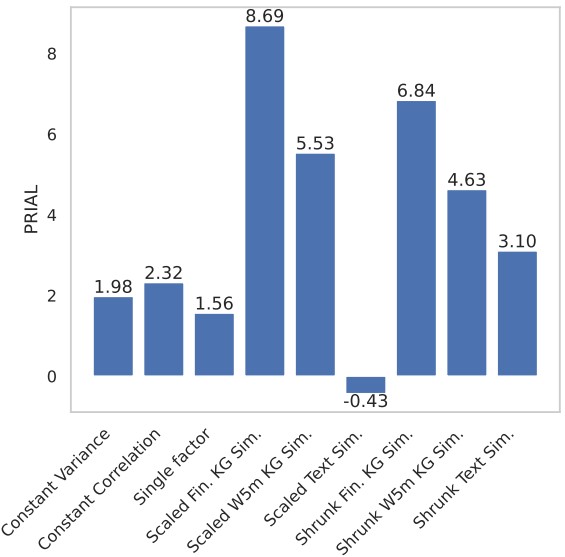

Figure 14: PRIAL for the NASDAQ-100 for 2007-2023

The improved consistency of the proposed estimators is highlighted in Figure 15. This shows that even scaled estimators showing a high PRIAL score (such as the scaled KG similarities) do not consistently perform better than other estimators. The PRIAL metric is an average metric and therefore sensitive to outliers. The scaled similarity methods are biased towards lower covariance values (see Table 1) leading to a very significant error reduction during volatility regime transition periods when the sample covariance is a poor estimator of future covariance (such as the low to high volatility transition during the coronavirus pandemic), but worse performance than other estimators overall.

The PRIL (defined in equation 25) allows visualization the contribution of each time window to the PRIAL. Figure 17 confirms the previous observation: the semantic shrinkage estimator outperforms its scaled similarity variant over the entire period, except for the transition into the high volatility events in 2020.

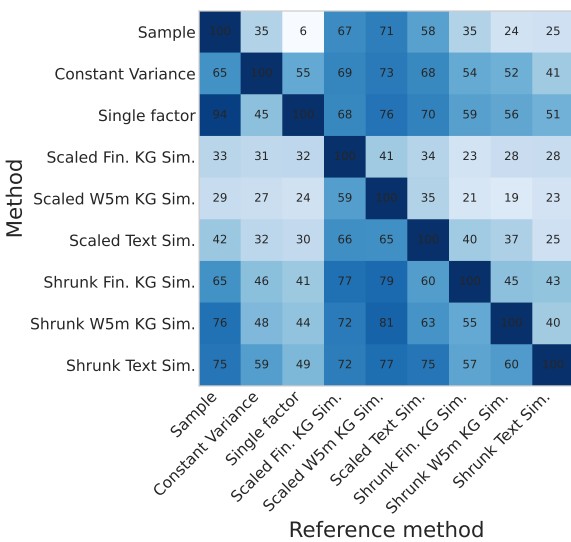

Figure 15: Percentage of rolling windows when a method (left) MSE is lower than a reference method (bottom) for the NASDAQ-100 index.

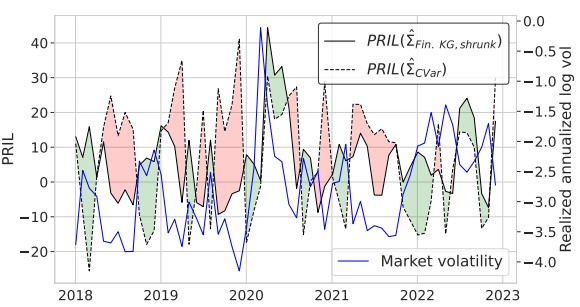

Figure 16: Relative performance of the Fin. KG similarity shrunk estimator to the constant variance estimator

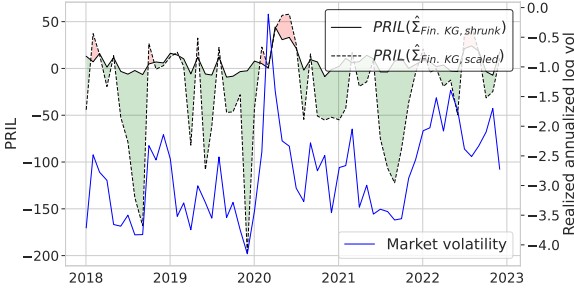

Figure 17: Relative performance of the Fin. KG similarity shrunk estimator to the KG similarity scaled estimator

A similar effect can be seen for the text-based estimators in Figures 18 to 19.

Similarly to results presented for the S&P 500 in 7, the degree of improvement from semantic shrinkage estimators for the covariance prediction shows a positive correlation with a measure of overall market volatility, as illustrated in Figure 20.

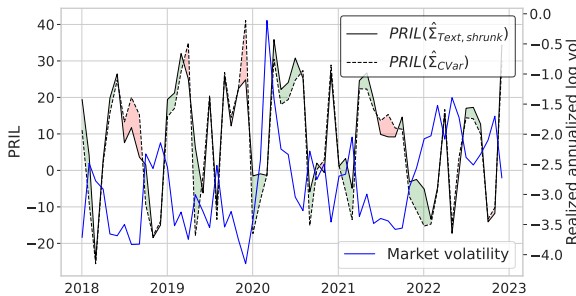

Figure 18: Relative performance of the Text similarity shrunk estimator to the constant variance estimator

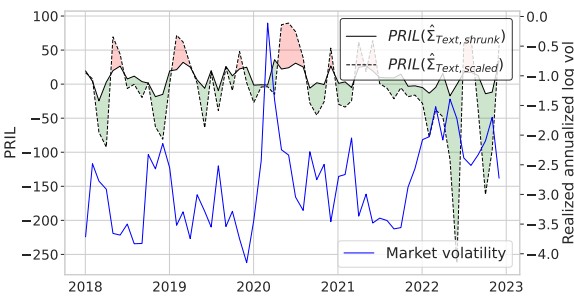

Figure 19: Relative performance of the Text similarity shrunk estimator to the Text similarity scaled estimator

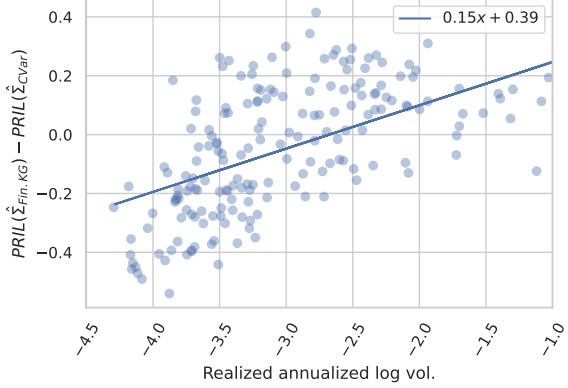

Figure 20: PRIL difference between KG and constant variance shrunk estimators for the NASDAQ-100

# D Experimental results for Nikkei 225

This appendix provides experimental results for the Nikkei 225 components. Figure 21 shows the PRIAL (defined in equation 24) over the entire experimental period. The PRIAL value for the scaled estimators is, unlike for the other two indices, higher than for their shrunk variant, with all semantic estimators significantly outperforming the estimators based on price only (constant variance, constant correlation and single factor shrinkage).

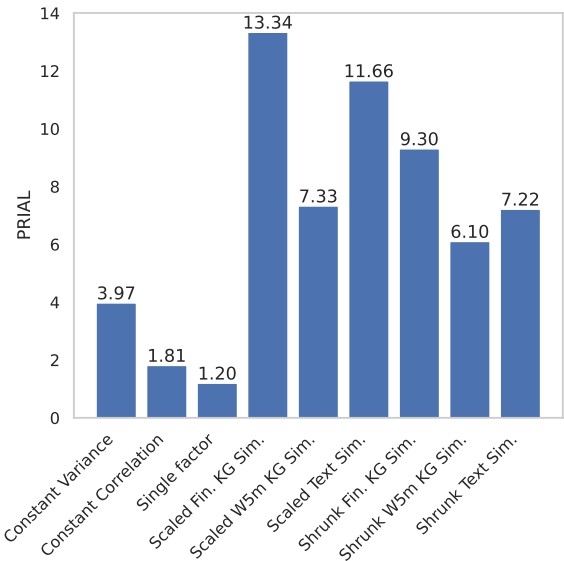

Figure 21: PRIAL for the NASDAQ-100 for 2007-2023

This result can be explained by the sensitivity of the PRIAL metric to outliers. Figure 22 shows that the improved PRIAL of the scaled estimators over their shrunk variant is not consistent: it can be seen that the scaled similarity estimators only outperform the price-based and the semantic shrinkage estimators 20 to 40% of the time (middle 3 rows). The semantic shrinkage estimators, on the other hand, tend to be consistently better than the other estimators (bottom 3 rows). This is also illustrated by the PRIL in Figures 24 and 24: the scaled similarity outperforms its shrunk variant only in the 2020 volatility transition period. The errors made by all estimators during this period are significantly higher than average, biasing the average PRIAL metrics.

Providing a reliable estimate of future covariance is a desirable feature and the proposed shrunk semantic similarity estimators show a consistent improvement over their scaled variants.

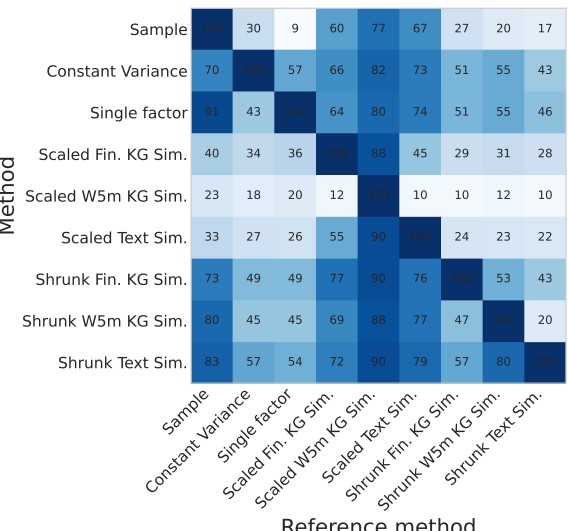

Figure 22: Percentage of rolling windows when a method (left) MSE is lower than a reference method (bottom) for the Nikkei 225 index.

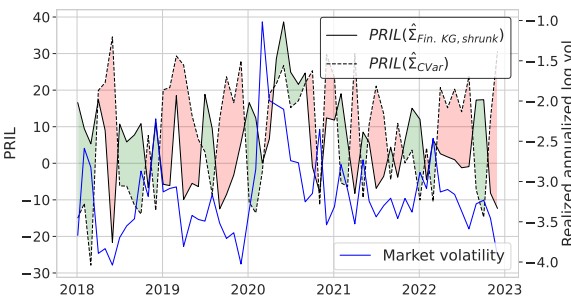

Figure 23: Relative performance of the Fin. KG similarity shrunk estimator to the constant variance estimator

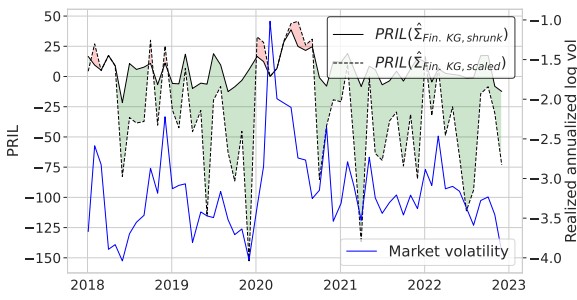

Figure 24: Relative performance of the Fin. KG similarity shrunk estimator to the KG similarity scaled estimator

# E   Semantic Models Hyperparameters

## E.1   Wikidata5m Knowledge Graph Embeddings

| Parameter | Value |
| --- | --- |
| Architecture | RotatE |
| | (Sun et al., 2019) |
| Dataset | Wikidata5m |
| | (Wang et al., 2021) |
| Dimension | 512 |
| Optimizer | SGD |
| Learning rate | 1e-2 |
| Weight decay | 0 |
| Margin | 6 |
| Batch size | 100k |
| Epochs | 1000 |
| Sample batch size | 2000 |
| # Negative/positive | 64 |
| Adversarial temp. | 0.2 |

Table 2: Wikidata5m KG Embeddings hyperparameters

## E.2   Text Embeddings

| Parameter | Value |
| --- | --- |
| Architecture | DistilBERT |
| | (Sanh et al., 2019) |
| Dataset | Natural Questions |
| | (Kwiatkowski et al., 2019) |
| Dimension | 768 |
| Hidden dim. | 3072 |
| # layers | 6 |
| # heads | 12 |
| Activation | GeLU |
| Loss | Multiple Negatives Ranking |
| Dropout | 0.1 |
| Cls. dropout | 0.2 |
| QA dropout | 0.1 |

Table 3: Text Embeddings hyperparameters

## E.3   Financial Knowledge Graph Embeddings

| Parameter | Value |
| --- | --- |
| Architecture | RotatE |
| | (Sun et al., 2019) |
| Dataset | Proprietary |
| Dimension | 128 |
| Optimizer | Adagrad |
| | (Duchi et al., 2011) |
| Learning rate | 1e-1 |
| Weight decay | 1e-7 |
| Margin | 19.9 |
| Batch size | 1024 |
| Epochs | 20 |
| Sample batch size | 512 |
| # Negative/positive | 2 |
| Adversarial temp. | 1.0 |

Table 4: Financial KG Embeddings hyperparameters