# OpenReview forum: "Semantic Similarity Covariance Matrix Shrinkage"
_EMNLP/2023/Conference — EMNLP 2023 Findings_

### Official Review · Reviewer_yKen · 2023-07-30

**Typos Grammar Style And Presentation Improvements:** 1. At the end of Introduction section…
**Soundness:** 3

**Excitement:**

4: Strong: This paper deepens the understanding of some phenomenon or lowers the barriers to an existing research direction.

**Paper Topic And Main Contributions:**

The estimation of covariance matrix of financial assets is a notoriously difficult problem given its high dimension. This paper proposes a novel approach for covariance matrix estimation by leveraging semantic similarity derived from textual descriptions or knowledge graphs. The authors use the semantic similarity as a shrinkage target rather than a direct estimator of the covariance, resulting in covariance estimators that combine both semantic similarity and recent price history. The effectiveness of the approach is demonstrated using historical S&P 500 data and compared with other covariance shrinkage methods.


**Questions For The Authors:**

(1).In line 516, it is claimed that all values are positive. However, "scaled text simu" is -1.57. Do I missing anything?
(2) again, there are 190 data points. I believe only the average values of PRIAL are presented in Figure 4. Why don't you present some other statistics, such as what is the percentage of data points that the semantic methods outperform?

**Reasons To Accept:**

1) The paper introduces a new methodology for covariance matrix estimation in portfolio optimization, which incorporates semantic similarity derived from textual descriptions or knowledge graphs. The linear shrinkage method was proposed by Stein in 1956. The innovation of this paper lies in using semantic information to estimate the shrink target matrix.
2) The experiments shows that various semantic methods(either DistilBERT or W5m KD) improves the estimation of the covariance matrix.


**Reasons To Reject:**

1) The exposition and clarity of the paper could be improved. Certain sections, such as the derivation of the shrinkage factor estimators, may be difficult for readers to follow without more detailed explanations or examples.

**Reproducibility:**

3: Could reproduce the results with some difficulty. The settings of parameters are underspecified or subjectively determined; the training/evaluation data are not widely available.

**Reviewer Confidence:**

4: Quite sure. I tried to check the important points carefully. It's unlikely, though conceivable, that I missed something that should affect my ratings.

---

> ### Author Rebuttal · Authors · 2023-08-25
>
> Thank you for the suggestions.
>
> A: **All PRIAL values positive** line 516: all semantic similarity *shrinkage* estimators improve upon the baseline (the negative value is for the raw similarity matrix that was used in the prior art). This section has been updated for clarity, thank you.
>
> B: **Evaluation**: a matrix showing the percentage of data points where a method outperforms other methods has been added. This effectively captures how often a method outperforms the others and eliminates the impact of outliers on the performance difference. This shows that our method not only improves on the average values (via PRIAL), but is also more consistent and reliable (especially when compared to the raw or scaled similarity matrix estimator used in the prior art). We have run extended statistical this analysis on the S&P 500 and additional indices (Nikkei 250, NASDAQ 100). Thank you for the suggestion.

---

### Official Review · Reviewer_NdPC · 2023-07-30

**Soundness:** 3

**Excitement:**

3: Ambivalent: It has merits (e.g., it reports state-of-the-art results, the idea is nice), but there are key weaknesses (e.g., it describes incremental work), and it can significantly benefit from another round of revision. However, I won't object to accepting it if my co-reviewers champion it.

**Paper Topic And Main Contributions:**

This paper provides a method for better estimating the covariance matrix of different stocks in a stock market. The method is based on shrinkage methods in statistics and the semantic embeddings of companies generated by existing language models and knowledge graphs. Experiments show that the proposed method achieves lower estimation error, and validate a few hypothese regarding how semantic similarities should be used in the context of portfolio optimization.

**Questions For The Authors:**

**A.** An important quantity in the evaluation of covariance estimation methods is the MSE loss funciton (4), which is based on the population covariance $\Sigma^*$. I couldn't find the definition or a description of population covariance in the paper. Though this is a standard concept in statistics, it is unclear what it refers to in the specific datasets in the experiments, or how its value is computed in the evaluation.

**B.** Line 422: While 98.3% of the stocks are mapped to a Wikipedia entry, how did you handle the rest 1.7%?

**C.** Line 495: The claim that higher dimension models have lower off-diagonal similarity values sounds premature. The models in Table 1 having different dimensions are different types of models, so the dimension is not the single variable here. You can train each type of model with different embedding dimensions. If all types of models exhibit the same pattern, the above claim will be valid then.

**D.** In Table 1, should the sample covariance matrix be scaled as well (making the diagonal entries be 1), for a fair comparison of the off-diagonal Frobenius norms?

**Reasons To Accept:**

The paper could belong to the "NLP Applications" track, and is of specific interest to readers in the FinTech community who apply NLP techniques. The paper is an exhibition of wider adaptability of embedding models than normally presented in the conference. The paper shows that the semantic similarity matrix cannot be used directly as the covariance matrix, but as a shrinkage target.

**Reasons To Reject:**

Overall, the paper would be more interesting to the EMNLP audience if it compares the embedding models in more detail and analyzes why one model has better performance than another one. The current writing uses language models and knowledge graphs as a black box and might be more suited to a FinTech conference.

The technical part is mostly sound, but I have a few questions listed below that need to be resolved. Also, the experiment stops at the estimation of covariance matrix, and the paper doesn't present real-world benefits of the methods such as in portfolio optimization. I would expect a more thorough set of experiments that shows the impact of better covariance estimation with the proposed method.

**Reproducibility:**

4: Could mostly reproduce the results, but there may be some variation because of sample variance or minor variations in their interpretation of the protocol or method.

**Reviewer Confidence:**

4: Quite sure. I tried to check the important points carefully. It's unlikely, though conceivable, that I missed something that should affect my ratings.

**Typos Grammar Style And Presentation Improvements:**

The authors could use more references in some of the writing, as sources for their claims. For example, lines 105-107: where does this claim come from, and why is it a shortcoming? (readers may guess but it's better to state clearly)

The section titles "Linear shrinkage" and "Semantic similarity shrinkage" look as if the proposed method was not linear shrinkage. It may be better to change the former to "Preliminaries".

The definitions of some notations are missing.
* Line 95: N and p were not defined until Line 207.
* Line 356: Where are the definitions of ti, tj?

---

> ### Author Rebuttal · Authors · 2023-08-25
>
> Thank you for the improvement suggestions.
>
> We believe this paper would be a valuable contribution to the NLP community for the following reasons:
> - While illustrated for finance applications, the approach is valid for all domains where highly dimensional covariance matrices must be estimated (e.g. medical imaging, sensors) and not restricted to a finance audience
> - The NLP community has a strong focus on creating state of the art semantic models, yet even recent work (2022 and later) has assumed that the resulting similarity matrices were valid estimators for the covariance matrix, which is incorrect as demonstrated by our work. We aim to grow awareness of these issues with this work.
> - Linear shrinkage is an effective and widely used method in quantitative finance for highly dimensional covariance matrix estimation, but is not widely known in the NLP field. We believe our contribution would benefit the NLP community by illustrating how an established method can be adapted for use with NLP semantic models.
>
> A: **Population covariance definition** The true population covariance risk is unobservable since the distribution of the random variables is unknown. The derivation of the optimum shrinkage factor uses asymptotic analysis. For evaluation purposes, we use the empirical risk via the realized sample covariance matrix over the test set as an estimate of the population risk. Further details have been added to the article for the definition of the loss and evaluation metric.
>
> B: **Stocks not found in Wikidata** The 1.7% remaining stocks were excluded from the experiment for both training and evaluation - note added
>
> C: **Dimensionality and Frobenius norm** The claim linked to the dimensionality of the embeddings will be removed to maintain focus, and replaced by the observation that the expected frobenius norm of the similarity matrix depends on the underlying semantic model (e.g. text or KG).
>
> D: **Scaling of the sample covariance** The sample covariance matrix is an unbiased estimator of the population covariance and should not be scaled. The aim of this table is to show other estimators are on the other hand biased (different than the sample covariance matrix) : It is a necessary condition for the estimators proposed to have a Frobenius norm close to the sample covariance matrix for them to be unbiased estimators. This point will be clarified in the camera ready version
>
> **Typos and grammar**:
> Lines 105-107: Linear shrinkage estimators built using price data can be seen as empirical Bayes estimators using a data dependent prior (using the same data used to evaluate the likelihood). Thank you - a clarification note and reference were added.
> Rest of suggestions has been implemented.

---

### Official Review · Reviewer_EMEZ · 2023-08-03

**Soundness:** 3

**Excitement:**

3: Ambivalent: It has merits (e.g., it reports state-of-the-art results, the idea is nice), but there are key weaknesses (e.g., it describes incremental work), and it can significantly benefit from another round of revision. However, I won't object to accepting it if my co-reviewers champion it.

**Paper Topic And Main Contributions:**

The paper presents an approach to improve covariance matrix estimation for portfolio optimization. Covariance matrix estimation often suffers from the curse of dimensionality. This is a common scenario in portfolio optimization, where portfolio managers choose between thousands of stocks using historical daily returns.

While existing linear covariance shrinkage methods are effective, they rely solely on historical price data and ignore company fundamental data. To overcome these limitations, the authors propose to extend the linear shrinkage framework to semantic similarity matrices. They utilize semantic similarity, derived from textual descriptions or knowledge graphs, as a shrinkage target for covariance matrix estimation. This approach leverages both semantic similarity and recent price history.

The paper examines and highlights the limitations of using semantic similarity to directly estimate the covariance matrix. It argues that while semantic models are effective at ranking similarities, their absolute similarity value is typically uncalibrated, dependent on the fraction of the hypersphere occupied by embeddings, and their expected cosine similarity value depends on the dimension of embeddings.

The authors propose three hypotheses:
1. Semantic similarity depends on the embedding dimension and cannot be used directly as a covariance matrix estimator.
2. Semantic similarity can be used as an effective regularization target for covariance matrix estimation.
3. The proposed shrinkage target includes fundamental information about random variables and is less sensitive to sudden changes in the volatility regime for those variables.

They validate these hypotheses using a dataset comprising historical members of the S&P 500 index over a period of 16 years and demonstrate that the semantic similarity-based models outperform the existing price-based shrinkage methods, especially during periods of high market volatility.

**Questions For The Authors:**

The authors should address the weaknesses mentioned in the above section.

**Reasons To Accept:**

1. The paper proposes an approach to improve covariance matrix estimation. It introduces a method that extends the linear shrinkage framework to semantic similarity matrices.

2. The proposed approach utilizes semantic similarity, derived from textual descriptions or knowledge graphs, as a shrinkage target for covariance matrix estimation. It leverages both semantic similarity and recent price history, unlike existing linear covariance shrinkage methods, which rely solely on historical price data.

3. The paper examines and highlights the limitations of using semantic similarity matrices directly as covariance matrix estimators.

4. The paper conducts an experimental evaluation of the proposed method using historical members of the S&P 500 index over a period of 16 years. The authors compare their approach with traditional ones under diverse market conditions and demonstrate its superiority, especially during periods of high market volatility.

**Reasons To Reject:**

1. The paper relies heavily on historical S&P 500 data for the experimental evaluation. More experimentation on other stock indexes and regions could further validate the approach.

2. The proposed similarity-based targets are built from static semantic information. They do not incorporate dynamic components, such as recent news data. The authors acknowledge this and suggest it as a direction for future work.

3. The input to the semantic models (text and knowledge graph triples) was gathered as of 2021 and 2022. For the 2007-2023 period backtest, there might be a risk of data leakage. However, the authors estimate this risk to be low.

4. Although the authors have mentioned the limitations in the paper, they should provide a more detailed plan on how they plan to address these drawbacks in their future work.

5. The authors should provide a detailed hyperparameter analysis.

6. The paper focuses on the portfolio optimization setting. Extensions to other financial applications utilizing covariance matrices could benefit the paper.

7. The Python Github Repository is REDACTED on Page 2. Also, some information is REDACTED on Page 6. The authors should mention why they are redacted and when they will be available.

**Reproducibility:**

3: Could reproduce the results with some difficulty. The settings of parameters are underspecified or subjectively determined; the training/evaluation data are not widely available.

**Reviewer Confidence:**

2: Willing to defend my evaluation, but it is fairly likely that I missed some details, didn't understand some central points, or can't be sure about the novelty of the work.

**Typos Grammar Style And Presentation Improvements:**

Figure 2 caption: "BICS industry sector" should be "BICS industry sectors" (plural)

---

> ### Author Rebuttal · Authors · 2023-08-25
>
> Thank you for the suggestions.
>
> 1: **Additional stock indices**: Experiments on Nikkei 250 and Nasdaq 100 have been performed (same full experiment set provided for S&P 500). Results align with S&P 500 findings and will be included in the camera ready paper.
>
> 2: **Static/Dynamic semantics** The proposed approach does not include dynamic semantic information, and future work could investigate news data as an additional semantic input. Text embeddings from news data (capturing company recent events) can be easily combined with the static embeddings (capturing company fundamentals) to compute either a single similarity matrix to be shrunk, or use techniques such as multiple targets shrinkage. Challenges with this approach will be the sampling of news data to be representative of the company context, lack of news for some companies and importance given to the static versus dynamic components. These suggestions will be added in the limitations section
>
> 3: **Data leakage**: The price data frequency is typically much higher than frequency of change of the fundamental data (description or KG triples): the company fundamental characteristics (e.g. industry sector) of the large capitalization investigated did not change significantly over the period considered. Furthermore, the analysis does not show a drop in performance before and after the snapshot were selected. For future work including dynamic data (e.g. news), it will be critical to ensure that only news released prior to an experiment time window are used, since news do have a significant impact on short term price data (and correlation) as illustrated by recent work referenced in our work. These considerations will be included in the camera ready paper.
>
> 5: **Hyperparameters**: The proposed method is a post-processing step to semantic models, rather than a new semantic modeling approach. Our shrinkage method’s only parameter (the shrinkage factor) is analytically calculated. Nevertheless the full set of hyperparameters for the training of the 2 KG embeddings and the sentence embedding model will be added to the appendix.
>
> 6: **Further applications**: The applicability of the approach to other domains using large scale covariance matrices will be illustrated in the introduction via a few additional examples (both finance applications: company replacement in a fund, value at risk or time series methods  and non-finance related: medical imaging and sensors).
>
> 7: **Redacted information** The information was redacted for anonymity purposes. The source of the knowledge graph and github repository will be un-redacted for the camera-ready version
>
> Grammar style and presentation improvements have been addressed

---

### Meta-Review · Area_Chair_3wxV · 2023-09-18

**Recommendation:** 4

**Metareview:**

The paper introduces a new methodology for covariance matrix estimation in portfolio optimization, which incorporates semantic similarity derived from textual descriptions or knowledge graphs. The innovation of this paper lies in using semantic information to estimate the shrink target matrix. Overall, this could be an interesting application paper.

---

### Decision · Program_Chairs · 2023-10-07

**Decision:**

Accept-Findings

**Comment:**

The paper introduces a new methodology for covariance matrix estimation in portfolio optimization, which incorporates semantic similarity derived from textual descriptions or knowledge graphs. The innovation of this paper lies in using semantic information to estimate the shrink target matrix. Overall, this could be an interesting application paper.